# Understanding the Size of the Feature Importance Disagreement Problem in Real-World Data

**Aniek F. Markus** [1]  **Egill A. Fridgeirsson** [1]  **Jan A. Kors** [1]  **Katia M.C. Verhamme** [1]  **Jenna M. Reps** [1,2]
**Peter R. Rijnbeek** [1]

## Abstract

Feature importance can be used to gain insight in prediction models. However, different feature importance methods might result in different generated explanations, which has recently been coined as the explanation disagreement problem. Little is known about the size of the disagreement problem in real-world data. Such disagreements are harmful in practice as conflicting explanations only make prediction models less transparent to end-users, which contradicts the main goal of these methods. Hence, it is important to empirically analyze and understand the feature importance disagreement problem in real-world data. We present a novel evaluation framework to measure the influence of different elements of data complexity on the size of the disagreement problem by modifying real-world data. We investigate the feature importance disagreement problem in two datasets from the Dutch general practitioners database IPCI and two open-source datasets.

## 1. Introduction

Model transparency or insight is important in the current state of AI maturity. This is especially true in high-stakes domains such as healthcare, where clinicians are held accountable for their actions and should be able to explain their decision making to both colleagues and patients (Tonekaboni et al., 2019). Hence, transparency is often needed for model adoption in clinical practice. Feature importance (FI) is a type of (post-hoc) explanation that is often asked for by both developers and end-users to give insight into why model predictions are made. FI methods rank or measure

the predictive power of features and can help users verify if the system works as intended or discover relevant features for the prediction task (Hase et al., 2021; Markus et al., 2021).

In the literature, numerous FI methods have been proposed to generate such explanations. These FI methods differ along several dimensions. First, local explanations provide a rationale for an individual prediction, whereas global explanations attempt to explain the model as a whole. Second, model-agnostic methods are suitable to explain any kind of model, whereas model-specific methods are only suitable for specific model classes. Finally, feature importance is sometimes used to refer to a property of the model (i.e. model-agnostic and model-specific methods), and sometimes to a property of the data generating process (i.e. model-free methods). In this work, we focus on model-agnostic methods to compute global FI.

There is a more general awareness for the need of a more formal evaluation of FI methods. Often, new methods are proposed without systematic comparison to existing methods. Various recent papers aimed to classify and unify methods according to their behavioral characteristics (Covert et al., 2021; Han et al., 2022). One of the main challenges to evaluate FI methods is the lack of a ground truth, which makes it impossible to say which method is preferred for the given problem setting. The underlying problem is the lack of consensus on a definition of what it means for a feature to be important (Hama et al., 2022; Mase et al., 2022). Despite this, feature importance is often used and presented as a uniform notion in practice. Practical guidance to motivate the choice between FI methods is still lacking.

It is increasingly recognized that there might be conflicting results between methods, i.e. generated explanations might differ in terms of the top features, feature ordering, and directions of feature contributions. This has been summarized as the explanation disagreement problem (Krishna et al., 2022). This phenomenon is not unexpected given that FI methods differ along several known dimensions (e.g. how features are removed, what model behavior is analyzed, and how importance is summarized) (Covert et al., 2021). However, such explanation disagreements are very harmful in

[1]Department of Medical Informatics, Erasmus University Medical Center, Rotterdam, The Netherlands [2]Janssen Research and Development, Raritan, New Jersey, United States. Correspondence to: Aniek Markus <a.markus@erasmusmc.nl>.

*Workshop on Interpretable ML in Healthcare at International Conference on Machine Learning (ICML)*, Honolulu, Hawaii, USA. 2023. Copyright 2023 by the author(s).

practice as conflicting feature importances only make the model non-transparent, while the aim of feature importance is to make models more transparent. Research is needed to investigate what causes this disagreement.

User studies by Krishna et al. (2022) found that the explanation disagreement problem is widely encountered (84% of interviewed participants experienced this in their day-to-day workflow). When developing clinical prediction models using routinely-collected health care data, such as electronic health record (EHR) or claims data, this problem is probably even larger due to the size, high-dimensionality, and sparsity of the data. For reliable explanation of clinical prediction models, it is important to empirically analyze and understand the FI disagreement problem in real-world data.

The contributions of our work are threefold:

- First, we extend the current body of empirical evidence by analysing the size of the disagreement problem in two moderate-sized real-world datasets from the Dutch general practitioners database IPCI (including 100 features as opposed to 20 features in previous work (Krishna et al., 2022)).

- Second, we propose a novel evaluation framework to understand what causes FI disagreement. Our framework measures the influence of different elements of data complexity on the size of the explanation disagreement problem in real-world data.

- Finally, we apply our proposed framework to two open-source datasets: COMPAS and German Credit.

The remainder of this paper is structured as follows. Section 2 covers related work. In Section 3, we discuss the proposed evaluation framework. In Section 4, we first discuss the datasets, then the setup of the experiments (model development and FI methods), and finally present the results. Section 5 concludes with main takeaways and directions for future work.

## 2. Related Work

### 2.1. Quantitative Evaluation of Feature Importance

We distinguish two main approaches to quantitatively evaluate explanation quality. First, *direct evaluation* assesses quality by comparing explanation to a known ground truth. This approach has the advantage that it is possible to make statements about which explanation is correct. However, in practice the underlying data generating process of real-world data is unknown and the training data only consists of features and outcomes, but does not contain (annotated) explanations. Hence, the ground truth is often lacking which makes evaluation of FI methods difficult. Second, *indirect evaluation* assesses quality by verifying whether ex-

planations satisfy desirable axioms or properties. Examples of such properties are stability/robustness (Agarwal et al., 2022a), predictive faithfulness (Petsiuk et al., 2018), monotonicity (Luss et al.), infidelity (Yeh et al., 2019), and remove-and-retrain (Hooker et al., 2019). The majority of work evaluating feature importance has focused on indirect evaluation, as developing an (artificial) ground truth is tedious and requires (strong) assumptions.

Several strategies have been used to overcome the lack of ground truth. A common strategy is to construct toy examples or generate synthetic data following a (simple) specified data generating process to ensure known relationships (Merrick & Taly, 2019; Johnsen et al., 2021; Verdinelli & Wasserman, 2023). In this case, researchers often design datasets with different levels of informative versus uninformative features, order of feature correlation, degree of non-linearity, and amount of noise. However, there is no guarantee that the trained classifiers do capture these relationships exactly. A related strategy introduced by Guidotti (2021) is to create ground truth by generating synthetic transparent classifiers. Their proposed SENECA generators allow to pursue systematic evaluation of local explainers for different types of data (tabular, images, text) and various problem settings. Instead of creating a synthetic dataset or model, Zhou et al. propose another strategy to create semi-natural datasets. They modify image and text datasets by using label reassignment (to reduce predictive power of features) and input manipulation (to introduce new features for the model to rely on). Finally, Krishna et al. (2022) circumvented the problem of a missing ground truth by comparing the explanation of one FI method against the explanations of all other methods. Here, the results of other FI methods are thus implicitly used as ground truth.

### 2.2. Metrics Evaluating Disagreement

The simplest type of evaluation is to assess whether a given FI method can identify the (known) set of important features. Other work has evaluated FI methods by comparing the resulting rankings or values. Rajbahadur et al. (2021) measured alignment with Kendall's Tau coefficient and Top-K overlap. Krishna et al. (2022) recently proposed six metrics to compare two lists of rankings: Feature Agreement (FA), Rank Agreement (RA), Sign Agreement (SA), Signed Rank Agreement (SRA), Rank Correlation (RC), and Pairwise Rank Agreement (PRA). Rengasamy et al. (2021) used three methods to evaluate two lists of values: Mean Absolute Error (MAE), Root Mean Square Error (RMSE), and Coefficient of Determination (R-squared).

### 2.3. Benchmarking Frameworks

Recently, several frameworks have been proposed to evaluate and benchmark explanations. First, Liu et al. (2021)

proposed XAI-Bench to evaluate the quality of post-hoc explanations using synthetic datasets with known ground-truth distributions. They argue that their released synthetic datasets can be configured to simulate real-world data. They use the ground truth for exact computation of evaluation metrics with respect to the original model. Carmichael & Scheirer (2021) proposed an evaluation framework based on the MatchEffects algorithm that enables comparison of explanations to ground truth derived from additive contributions of features in the model. Their algorithm does not consider a one-to-one matching of effects, but instead matches subsets of effects to account for possible interaction effects. More recently, Agarwal et al. (2022b) introduced the open-source benchmarking framework OpenXAI. This initiative aims to gain insight in empirical performance of FI methods for different datasets and models by reporting predictive faithfulness, stability, and fairness in a leaderboard. This approach relies on indirect evaluation. None of the aforementioned frameworks can be used to understand FI disagreement when no ground truth exists.

In this work, we propose a novel evaluation framework to measure explanation disagreement that allows for targeted experiments (i.e. controlling the problem setting) on real-world data using direct evaluation. This builds on the work of Krishna et al. (2022) who coined the explanation disagreement problem with the help of user studies and empirically analyzed the size of the problem in small datasets. Our work extends their empirical analysis to moderate datasets and provides insight on the size of the FI disagreement problem in relation to various elements of data complexity.

## 3. Understanding Feature Importance Disagreement using Real-World Data Modification

Model performance can be evaluated in real-world data by using a set of observations for which the ground truth is known and that has not been used for development. However, for explanations such ground truth does not exist in real-world data. For reliability of presented explanations, it is desirable that different FI methods agree on the given explanation. More agreement leads to more reliable explanations as the given explanation is less dependent on the (often subjectively) chosen FI method. The complexity of data is known to influence the behavior of FI methods. For example, correlated features can lead to non-zero importance of features that are irrelevant for the model or spread feature importance (Hooker et al., 2021; Rajbahadur et al., 2021; Verdinelli & Wasserman, 2023), and additional (uninformative) features can decrease the ability of FI explanations to comply with a linear ground truth explanation (Guidotti, 2021). Therefore, we propose a novel evaluation framework to measure the influence of different elements of data complexity on the size of the explanation disagreement problem in real-world data.

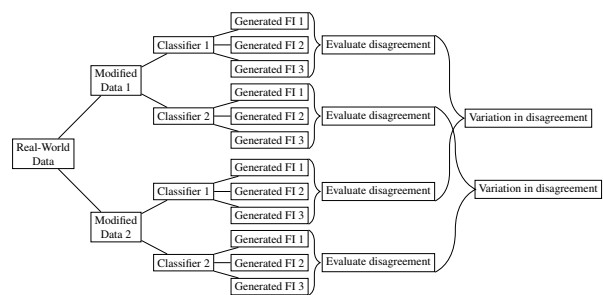

*Figure 1.* Proposed evaluation framework: for data complexity element of interest, modify the original real-world data to create new modified datasets with lower and higher levels of complexity, train classifiers and compute feature importances, measure the disagreement between the generated explanations.

Figure 1 shows the proposed framework. Evaluation using the framework proceeds as follows. First, specify which elements of data complexity are of interest. Second, adapt the original real-world data according to the proposed data modifications to create new datasets with lower and higher levels of complexity (see Section 3.1). Third, train classifiers and compute importance for each modified dataset. Finally, evaluate the disagreement between the generated explanations using various metrics (see Section 3.2).

There are several advantages of using real-world data instead of synthetic data for evaluation (Gentzel et al., 2019). First, the real-world data contains the complex relations and possible unknown influences that any synthetic data generation process most likely fails to capture (usually based on simplified researcher assumptions). Second, there are less degrees-of-freedom for the researcher which reduces the risk of (unintentional) bias and strengthens the validity of the results. Finally, using real-world data might be the most effective – if not the only – way to demonstrate reliability and limitations of explanations to end-users. A commonly mentioned advantage of synthetic data, however, is the ability to control all attributes of the dataset to perform targeted experiments in controlled settings (Liu et al., 2021). We exploit the advantages of both types of data by modifying real-world data to create semi-natural datasets.

### 3.1. Elements of Data Complexity

In this work, we consider the following data complexity elements of interest:

1. *Number of features.* The set of available candidate features might vary in size depending on the data collected in each database (e.g. only diagnoses and prescriptions, or also clinical measurements). The complexity of the generated explanations is highly dependent on the di-

mensionality of the input data. Higher dimensional input data leads to larger models and increases the complexity of the resulting explanation, both in terms of computation time as well as for human interpretation.

2. *Number of observations*. The available sample size might vary hugely across prediction tasks (depending on the target cohort) and databases in practice. Although the number of observations might be less influential for explanations than number of features, it has been shown that sample size impacts the complexity of the resulting prediction models (i.e. more observations, larger models) (John et al., 2022). Furthermore, for generating explanations, the sample size might determine whether it is possible to estimate (complex) conditional distributions or whether it is better to rely on (simpler) marginal distributions (Chen et al., 2022).

3. *Number of outcomes*. As previous work by John et al. (2022) showed that the relation between sample size and model complexity was primarily driven by the number of outcome events (i.e. less outcomes, smaller models), we next investigate variations in number of events. This mimics the situation where in practice outcomes might be harder to observe than non-outcomes (e.g. for rare diseases or expensive laboratory tests).

4. *Feature correlation*. In clinical data, correlations between features are common and often strong. When a patient is diagnosed with a certain disease, this patient most likely underwent several standard tests to confirm this (or to rule out other diseases) after which the patient starts the recommended treatment. Hence, certain combinations of diagnoses, prescriptions, and clinical measurements typically co-occur resulting in a high correlation between several candidate features. FI methods have different ways of dealing with feature dependencies; some assume independence between features (e.g. Permutation FI, marginal Shapley values), whereas others model the dependencies (e.g. SAGE conditional). High correlation between features complicates the interpretation of feature importance and might lead to incorrect conclusions about the relevance of features (Verdinelli & Wasserman, 2023).

5. *Prevalence of features*. A typical characteristic of EHR data is the sparsity as candidate features typically indicate the presence of diagnoses or prescriptions (which are most often absent). We do not know of any other work investigating feature importance in relation to feature sparseness, but are interested to investigate whether the frequency of binary features influences explanation disagreement.

For each element of data complexity, we suggest a data modification in Table 1. We reuse the previously modified data for increasingly severe modifications (e.g. the set of 10 candidate features is selected from the set of 25 candidate features).

### 3.2. Metrics

We measure disagreement of FI methods with a combination of ranking-based (i.e. measuring similarity between two lists of feature rankings) and value-based (i.e. measuring the difference between two lists of (normalized) feature importances) evaluation metrics. Although consistency of FI rankings was found to be most important (Krishna et al., 2022), we argue ultimately it is desirable for FI methods to also align on the relative importance of features (i.e. feature values).

Given a set of $|D|$ candidate features $\kappa = \{1, ..., D\}$, let $F_{K,m} \subseteq \kappa$ denote the subset of top-$K$ features selected by FI method $m$ (we use $K = 5$). Furthermore, let $V_{d,m} \in \mathbb{R}$ be the value, $R_{d,m} \in \mathbb{Z}^+$ the rank and $S_{d,m} \in \{+, -\}$ the sign of feature $d \in \kappa$. For any combination of FI methods $u$ and $v$ we quantify disagreement using different types of metrics from earlier work (Krishna et al., 2022; Rajbahadur et al., 2021; Rengasamy et al., 2021).

**Top-K feature metrics:**

- Feature Agreement (FA) = $\frac{|F_{K,u} \cap F_{K,v}|}{K}$

- Sign Agreement (SA) = $\sum_{d \in \{F_{K,u} \cap F_{K,v}\}} \frac{\mathbb{1}[S_{d,u} == S_{d,v}]}{K}$

- Signed Rank Agreement (SRA) = $\sum_{d \in \{F_{K,u} \cap F_{K,v}\}} \frac{\mathbb{1}[R_{d,u} == R_{d,v} \& S_{d,u} == S_{d,v}]}{K}$

**Feature ordering metrics:**

- Kendall's Rank Correlation (KRC) = $\frac{2}{D(D-1)} \sum_{\{c,d\} \in \kappa, c < d} sgn(R_{c,u} - R_{c,v}) sgn(R_{d,u} - R_{d,v})$

**Feature value metrics:**

- Mean Absolute Error (MAE) = $\sum_{d \in \kappa} \frac{|V_{d,u} - V_{d,v}|}{D}$

- Root Mean Squared Error (RMSE) = $\sqrt{\sum_{d=1}^{D} \frac{(V_{d,u} - V_{d,v})^2}{D}}$

For ease of interpretation, we scale all metrics such that their values range between 0 and 1 (using min-max normalization) with values closer to 1 indicating more agreement (i.e. MAE and RMSE are calculated as 1-value).

*Table 1.* Proposed data modifications.

| ELEMENT OF DATA COMPLEXITY | MODIFICATION OF REAL-WORLD DATA | VALUES ($\phi$) |
|---|---|---|
| NUMBER OF FEATURES | SELECT RANDOM SUBSET OF $\phi$ CANDIDATE FEATURES | 50, 25, 15, 10, 5 |
| NUMBER OF OBSERVATIONS | SELECT RANDOM SAMPLE OF $\phi$ OBSERVATIONS (WITHOUT REPLACEMENT) | 5000, 1500, 750, 500 |
| NUMBER OF OUTCOMES/EVENTS | UNDERSAMPLE OUTCOMES; RANDOMLY SELECT A MAXIMUM OF $\phi$ OUTCOMES WHILE KEEPING NON-OUTCOMES | 1000, 500, 250, 100 |
| FEATURE CORRELATION | REDUCE CORRELATION BY REMOVING ONE CANDIDATE FEATURE OF EACH FEATURE PAIR WITH MUTUAL SPEARMAN CORRELATION ABOVE THRESHOLD $\phi$ | 0.9, 0.7, 0.5, 0.3 |
| PREVALENCE OF FEATURES | REDUCE PREVALENCE BY RANDOMLY REMOVING $\phi\%$ OF ORIGINAL RECORDS (X=1) FOR EACH BINARY FEATURE | 5, 15, 25 |

## 4. Experiments

We carried out experiments using the evaluation framework outlined in Section 3. For the experiments we used four datasets, two machine learning algorithms, and seven FI methods. We first describe the included datasets (Section 4.1). Next, we discuss the experimental setup (Section 4.2) and the main findings (Section 4.3).

### 4.1. Datasets

For this study we used data from the Dutch Integrated Primary Care Information (IPCI) database (de Ridder et al., 2022). The IPCI data contains longitudinal, routinely-collected health care data from computer-based patient records of around 350 general practitioner (GP) practices throughout the Netherlands. The IPCI database has been mapped to the Observational Medical Outcomes Partnership Common Data Model (OMOP CDM), which enables standardized extraction and analysis of health care data (Overhage et al., 2012). The number of active patients in this dataset is 1.4 million, which comprises 8.1% of the Dutch population of 17 million.

We developed prediction models for two clinically relevant prediction tasks:

- Among adult patients newly diagnosed with chronic obstructive pulmonary disease (COPD), which patients will die in two years?

- Among newly diagnosed adult type 2 diabetes mellitus (T2DM) patients, which patients will be diagnosed with a cardiovascular disease (CVD) in five years?

Collected data includes patient demographics, information about contacts with GPs, symptoms, diagnoses, laboratory and clinical measurements, prescriptions, and information on use of secondary care. This study was approved by the IPCI Governance Board (number 03/2023).

As the IPCI database is not publicly accessible because of data privacy concerns (as commonly the case for routinely-collected health care data), we opted for a combination of datasets from the IPCI database and open-source data for benchmarking. For the latter we used the Correctional Offender Management Profiling for Alternative Sanctions (COMPAS) (Agarwal et al., 2022b) and German Credit datasets (UCI Machine Learning Repository).

For an overview of datasets used in this study see Table 2.

### 4.2. Experimental Setup

For each dataset, we trained prediction models using different ML algorithms using a 75%-25% train-test split. Let $f : X \rightarrow Y$ be a model that maps a set of input features $X$ to outcomes $Y$, where $x_i \in \mathbb{R}^D$ is a D-dimensional vector of features and $y_i \in [0, 1]$ a binary variable indicating the presence or absence of the outcome of interest. We investigated the following two model algorithms: L1 regularized logistic regression (LASSO) and neural network (NN). For each trained model $f$, we then determined global FI. We investigated the following model-agnostic FI methods:

- Permutation feature importance (Fisher et al., 2019):

$$PFI_d = \frac{1}{N} \sum_{i=1}^{N} L(y_i, \ f(x_i^{d, \ perm})) - L(y_i, f(x_i))$$

- Leave one covariate out (Lei et al., 2018):

$$LOCO_d = \frac{1}{N} \sum_{i=1}^{N} L(y_i, \ f(x_i)) - L(y_i, \ f^{-d}(x_i^{-d}))$$

*Table 2.* Details of open-source and IPCI datasets included in study.

| DATASET | TARGET POPULATION | OUTCOME OF INTEREST (TIME-AT-RISK) | NUMBER OF FEATURES | NUMBER OF OBSERVATIONS | OUTCOME RATE |
|---------|-------------------|-------------------------------------|--------------------|-------------------------|---------------|
| COMPAS | CRIMINAL DEFENDANTS | REOFFENSE (WITHIN 2 YEARS) | 7 | 6172 | 45.5% |
| GERMAN CREDIT | LOAN APPLICANTS | FAILING TO REPAY LOAN | 24 | 1000 | 30.0% |
| MORTALITY IN COPD | ADULT PATIENTS WITH NEW COPD DIAGNOSIS | ALL-CAUSE MORTALITY (WITHIN 2 YEARS) | 100 | 11145 | 9.4% |
| CVD IN T2DM | ADULT PATIENTS WITH NEW T2DM DIAGNOSIS | HEART FAILURE OR STROKE (WITHIN 5 YEARS) | 100 | 21494 | 8.7% |

- KernelSHAP (Lundberg & Lee, 2017):

$$\min_{\phi_1,...\phi_D} \sum_{S \subseteq \kappa} \frac{D-1}{\binom{D}{|S|}|S|(D-|S|)} (\sum_{d \in S} \phi_d - u(S))^2,$$

with $u(S) = \frac{1}{|B|} \sum_{x_b \in B} f(x_b)$ where $B$ is the background sample.

- SAGE (Covert et al., 2020):

$$\min_{\phi_1,...\phi_D} \sum_{S \subseteq \kappa} \frac{D-1}{\binom{D}{|S|}|S|(D-|S|)} (\sum_{d \in S} \phi_d - u(S))^2,$$

where $u(S) = -\mathbb{E}[L(\mathbb{E}[f(X)], Y)]$ for marginal and $u(S) = -\mathbb{E}[L(\mathbb{E}[f(X)|X^S], Y)]$ for conditional SAGE.

For PFI and LOCO we measured model loss $L$ by the area under the receiver operator curve (AUC) and mean squared error (MSE). For SAGE we used cross entropy loss $L$.

In the first set of experiments, we assessed baseline disagreement in the open-source and IPCI data. For feasibility of the experiments, we used a reduced real-world dataset with 100 covariates. Although developed prediction models might in practice be even larger, this is already much larger than previous studies and not all methods are computationally feasible on data with a higher number of features. In the second set of experiments, we examined how different elements of data complexity affect the disagreement between methods by transforming the real-world data as proposed in Section 3. For each combination of dataset, proposed modification, model algorithm, and FI method, we conducted five experimental runs to ensure that the results are stable and reliable.

For more details on the models and FI methods we refer to Appendix A. The full code to run the experiments (including the open-source datasets) can be found on GitHub: https://github.com/AniekMarkus/FIDisagreement.

### 4.3. Results and Insights

In this Section, we highlight the main findings. The remaining results can be found in Appendix B.

#### 4.3.1. SIZE OF DISAGREEMENT IN REAL-WORLD DATA

Figure 2 shows the disagreement between FI methods for LASSO predicting *Mortality in COPD* in IPCI data. Light cells indicate disagreement, whereas darker (blue) cells indicate more agreement. We examine disagreement using different metrics. Feature Agreement and Sign Agreement (Figure 2(a)-2(b)) show that especially SAGE-C, KernelSHAP, and LOCO MSE have high disagreement and very distinct feature importances compared to the remaining FI methods (and each other). We observe that agreement across all FI methods is much lower for Signed Rank Agreement (Figure 2(c)). This means a similar combination of rank and sign is rare and does not occur for approximately half the combinations of methods.

Across the top-5 feature metrics, we can find pairs of FI methods that consistently show strong agreement or disagreement. For example, the two PFI measures (AUC and MSE) are closely related, the two LOCO measures (AUC and MSE) much less. Similarly, LOCO AUC and PFI AUC show much higher agreement than LOCO MSE and PFI MSE. For feature ordering and value metrics, KernelSHAP and LOCO MSE show more agreement with other FI methods than SAGE-C. Results for predicting *CVD in T2DM* in IPCI data show slightly higher agreement with similar disagreement patterns across metrics (see Figure B.1). In the remainder of this work, we report average values across groups of metrics as individual metrics showed very similar patterns.

Figure 3 shows FI disagreement for LASSO versus NN predicting *CVD in T2DM* in IPCI data. We find higher disagreement for top-5 features for LASSO compared to NN. This suggests we observe more agreement between

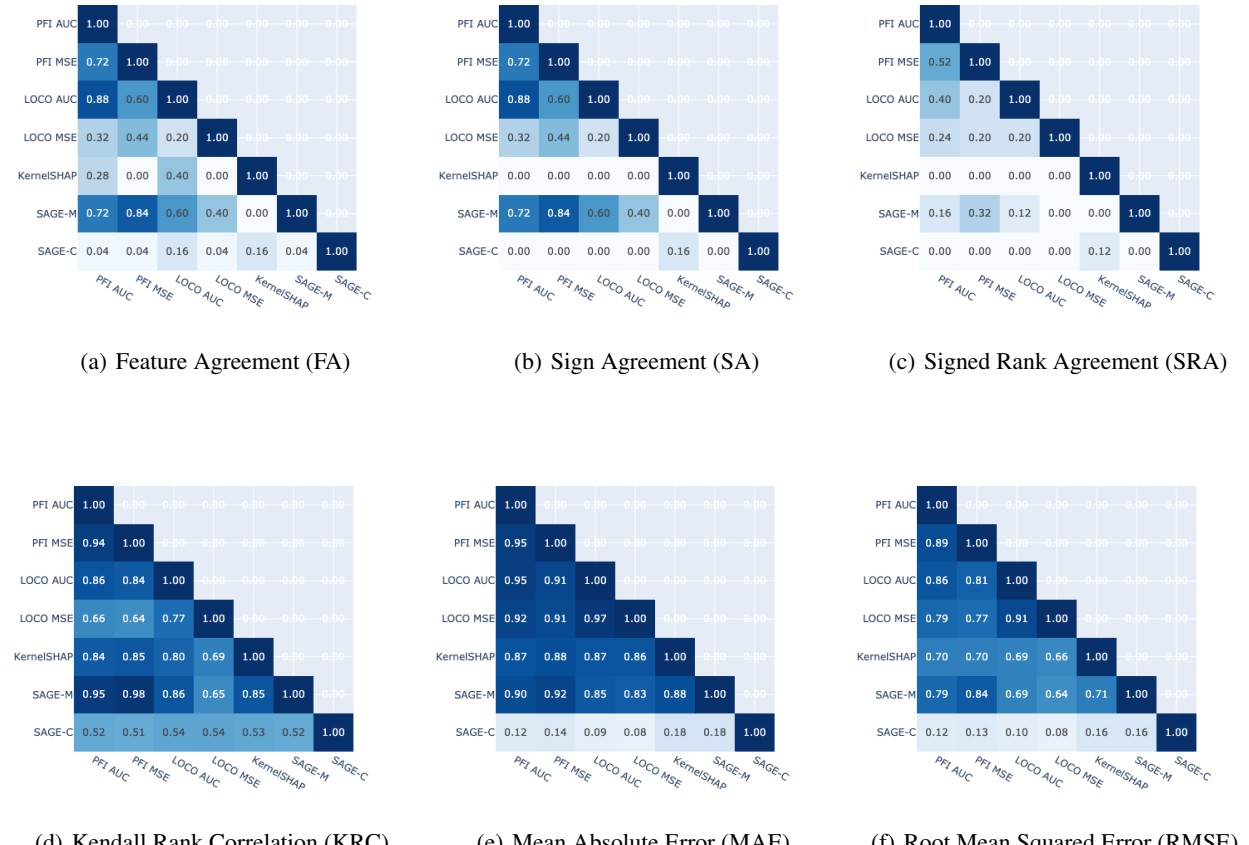

(a) Feature Agreement (FA)  (b) Sign Agreement (SA)  (c) Signed Rank Agreement (SRA)

(d) Kendall Rank Correlation (KRC)  (e) Mean Absolute Error (MAE)  (f) Root Mean Squared Error (RMSE)

*Figure 2.* Feature importance disagreement for LASSO predicting Mortality in COPD (IPCI data). Disagreement for all six metrics is averaged over five repeats. Values closer to 1 indicate more agreement.

all FI models for a simple compared to a more complex classifier. Similar patterns are observed for other metrics (Figure B.2) and the other IPCI dataset (Figure B.3).

Finally, compared to the moderate-sized real-world datasets, we find overall higher agreement in the smaller open-source datasets *COMPAS* and *German Credit*. This is especially true for COMPAS which only has 7 features. We again find agreement is much higher for LASSO (Figure B.4) than for NN (Figure B.5).

### 4.3.2. UNDERSTANDING THE DISAGREEMENT PROBLEM IN REAL-WORLD DATA

Figure 4 shows the disagreement between FI methods for varying levels of complexity as proposed in Section 3 for NN on the *German Credit* data. These results indicate the number of features has the largest influence on the overall level of agreement. Furthermore, the results show a small decrease in the overall level of agreement for increasing levels of complexity for the number of observations, feature

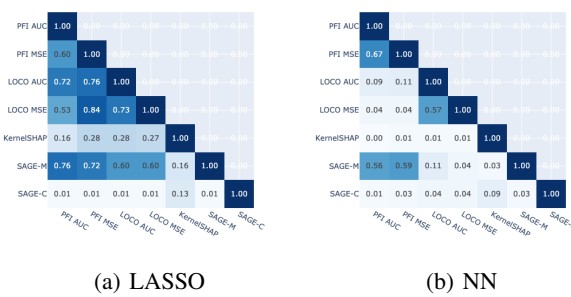

(a) LASSO  (b) NN

*Figure 3.* Feature importance disagreement for LASSO and NN predicting CVD in T2DM (IPCI data). Disagreement is measured using the average over top-5 feature metrics (FA, SA, SRA) over five repeats. Values closer to 1 indicate more agreement.

correlation, and prevalence of features, which may also be caused by a decrease in model complexity (i.e. reduced number of features). Even though the changes are relatively small, this is important as the models developed for the

modified data might not necessarily perform worse. Results from LASSO (Figure B.6) and the COMPAS dataset (Figure B.7-B.8) also show minor improvements for some elements of complexity.

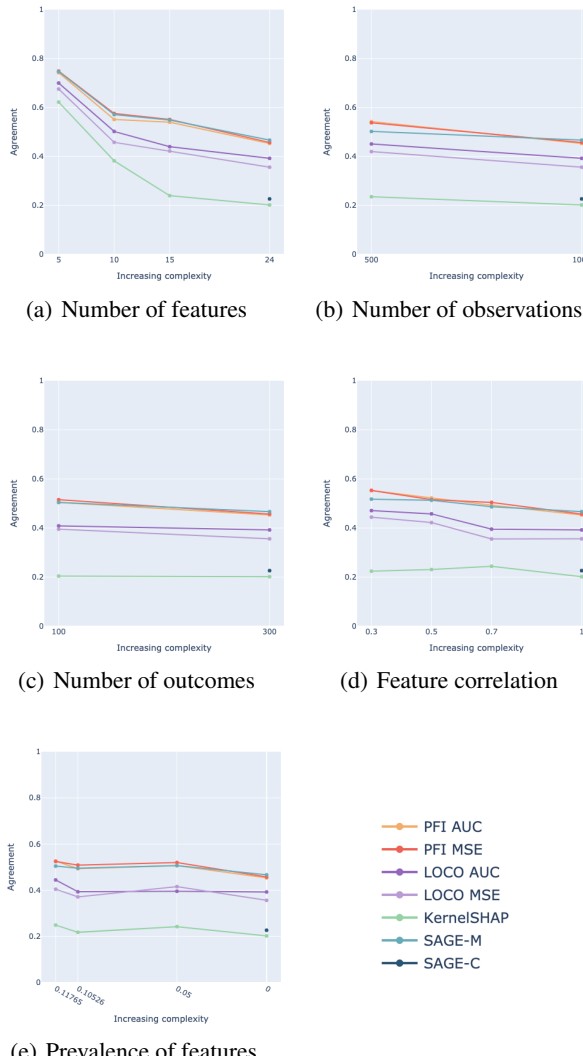

(a) Number of features

(b) Number of observations

(c) Number of outcomes

(d) Feature correlation

(e) Prevalence of features

*Figure 4.* Understanding size of feature importance disagreement for varying elements of data complexity for the NN model for German Credit (open-source data). Disagreement is measured using the average over Top-5 feature metrics (FA, SA, SRA) over five repeats. Values closer to 1 indicate more agreement.

## 5. Discussion and Conclusion

In this work, we aimed to better understand when explanations of various FI methods agree in real-world data. We show the FI disagreement problem is larger than in empirical evaluation with two small datasets. Furthermore, the problem is largest for more complex models (such as NN) that benefit most from additional explanations to improve

transparency. Our results show minor changes in overall disagreement when modifying elements of data complexity in real-world data. This should be explored further to investigate if there are differences between pairs of FI methods and whether this can be achieved without sacrificing model performance.

The proposed evaluation framework can directly be applied to other (tabular) real-world datasets, model algorithms, or FI methods. The main bottleneck to increase the scale of evaluation (e.g. to larger and more data sources across the OMOP CDM network) is the expensive computation time of some FI methods for high-dimensional data (e.g. KernelSHAP, SAGE marginal/conditional). This also limited the feasibility to run the second set of experiments on the IPCI data. Other limitations of our experiments include that we used limited sample size parameters for some FI methods to lower the computational burden (e.g. for KernelSHAP, SAGE).

Our evaluation framework could be extended to add other elements of data complexity (e.g. different levels of informative versus uninformative features, degree of non-linearity, presence of interactions) or to study interactions between elements of data complexity. Moreover, a similar strategy to modify real-world data can be used for other types of post-hoc explanations, such as evaluating properties of different counterfactual generation approaches.

For end-users of explanations, the insights obtained by using our proposed framework can help to understand limitations of explanations by quantifying differences between FI methods for various problem settings. Understanding limitations of explanations is crucial for interpretation with an appropriate level of trust. For researchers and developers, this framework can help to understand which pairs of FI methods (dis)agree in different problem settings. The question of how to reliably explain prediction models with feature importance in practice - given a certain goal of explanation and problem setting - is still unanswered. Understanding factors influencing the size of the disagreement problem is an important first step to be able to create guidance on which methods are more appropriate for which goal (e.g. to understand model decisions or to simplify models) and for different kinds of problem settings (e.g. high-dimensional and correlated EHR data). The next step is to investigate solutions for consensus feature importance when the problem setting is prone to disagreement. Possible solutions to explore include using grouped feature importance or heuristics to choose between methods based on data characteristics.

## Acknowledgements

This project has received support from the European Health Data and Evidence Network (EHDEN) project. EHDEN

received funding from the Innovative Medicines Initiative 2 Joint Undertaking (JU) under grant agreement No 806968. The JU receives support from the European Union's Horizon 2020 research and innovation programme and EFPIA.

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

# A. Experiment details and hyperparameters

*Table 3.* Hyperparameters model algorithms

| METHOD | SHORT NAME | IMPLEMENTATION | HYPERPARAMETERS |
|---|---|---|---|
| L1-REGULARIZED LOGISTIC REGRESSION | LASSO | PYTHON LIBRARY SCIKIT-LEARN | C=0.1 |
| DENSELY-CONNECTED FEED-FORWARD NEURAL NETWORK | NN | PYTHON LIBRARY TORCH | LAYERS = 4
NEURONS PER LAYER = (16, 32, 16, 1)
ACTIVATION = RELU, EXCEPT OUTPUT = SIGMOID
BATCH SIZE = 100
LEARNING RATE = 0.01 |

*Table 4.* Hyperparameters feature importance methods

| METHOD | SHORT NAME | IMPLEMENTATION | HYPERPARAMETERS |
|---|---|---|---|
| PERMUTATION FEATURE IMPORTANCE | PFI | CUSTOM CODE | REPEAT = 10
SCORING = AUC, MSE |
| LEAVE-ONE-COVARIATE OUT | LOCO | CUSTOM CODE | SCORING = AUC, MSE |
| KERNELSHAP | KERNELSHAP | PYTHON LIBRARY SHAP | SAMPLES=1000 |
| SAGE MARGINAL | SAGE-M | PYTHON LIBRARY SAGE-IMPORTANCE | PERMUTATIONS=1000
DETECTCONVERGENCE=TRUE |
| SAGE CONDITIONAL | SAGE-C | PYTHON LIBRARY SAGE-IMPORTANCE | PERMUTATIONS=1000
DETECTCONVERGENCE=TRUE
LAYERS = 4
NEURONS PER LAYER = (2*D, 64, 64, 2)
ACTIVATION = ELU
BATCH SIZE = 64
LEARNING RATE = 0.001 |

# B. Additional results

*Table 5.* Model performance for different algorithms on original datasets as measured by area under the receiver operator curve (AUC)

| DATASET | LASSO | NN |
|---|---|---|
| COMPAS | 0.746 | 0.692 |
| GERMAN CREDIT | 0.806 | 0.839 |
| MORTALITY IN COPD | 0.763 | 0.753 |
| CVD IN T2DM | 0.767 | 0.732 |

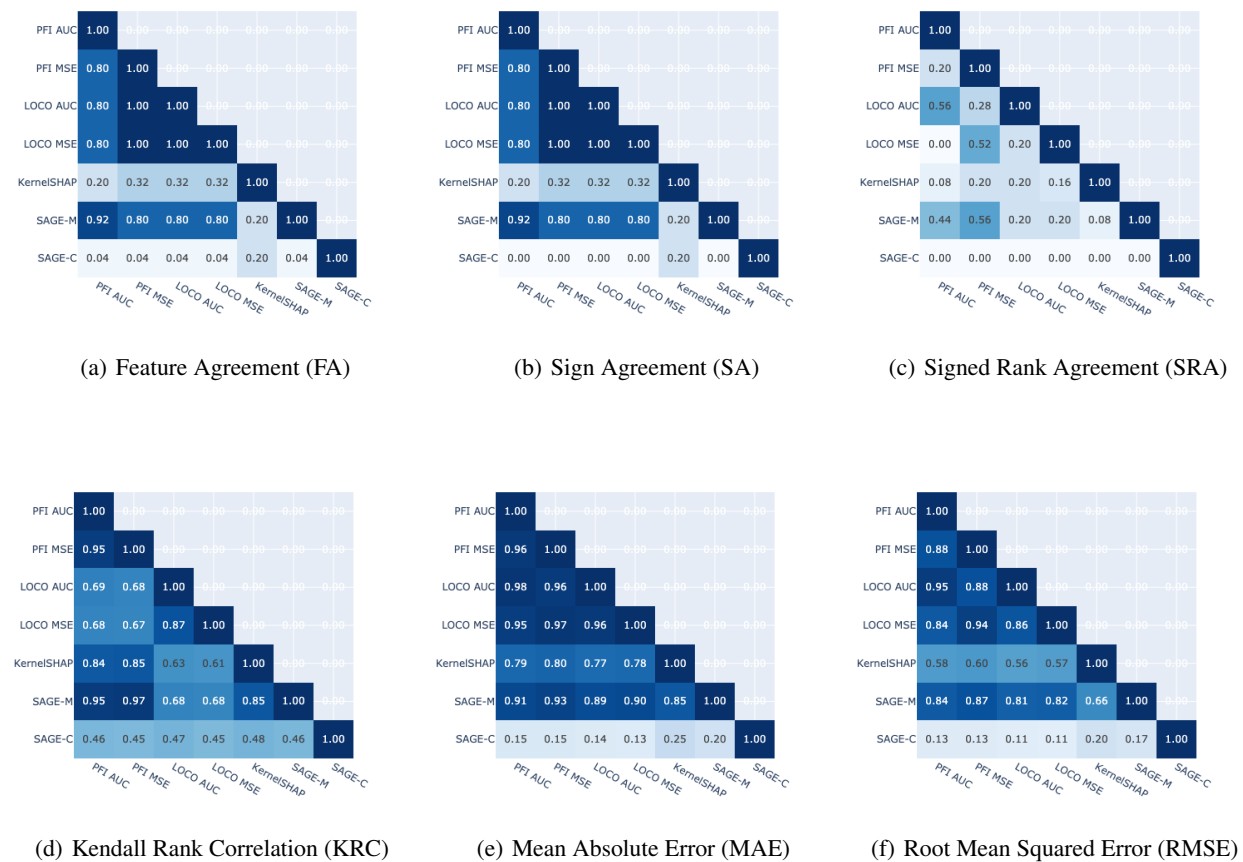

(a) Feature Agreement (FA)

(b) Sign Agreement (SA)

(c) Signed Rank Agreement (SRA)

(d) Kendall Rank Correlation (KRC)

(e) Mean Absolute Error (MAE)

(f) Root Mean Squared Error (RMSE)

*Figure B.1.* Feature importance disagreement for LASSO predicting CVD in T2DM (IPCI data). Disagreement for all six metrics is averaged over five repeats. Values closer to 1 indicate more agreement.

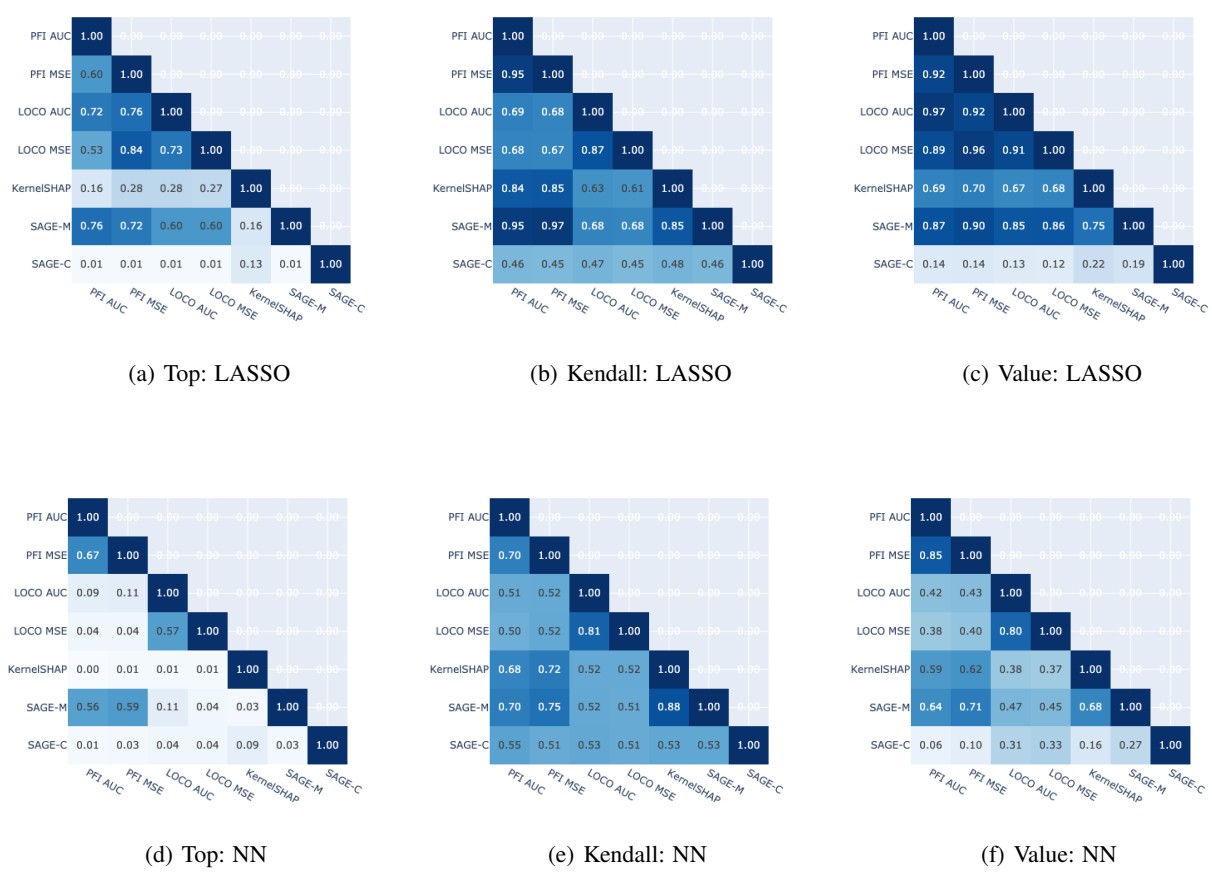

(a) Top: LASSO        (b) Kendall: LASSO        (c) Value: LASSO

(d) Top: NN        (e) Kendall: NN        (f) Value: NN

*Figure B.2.* Feature importance disagreement for LASSO and NN predicting CVD in T2DM (IPCI data). Disagreement is measured using the average over Top-5 feature, Feature ordering, and Feature value metrics over five repeats. Values closer to 1 indicate more agreement.

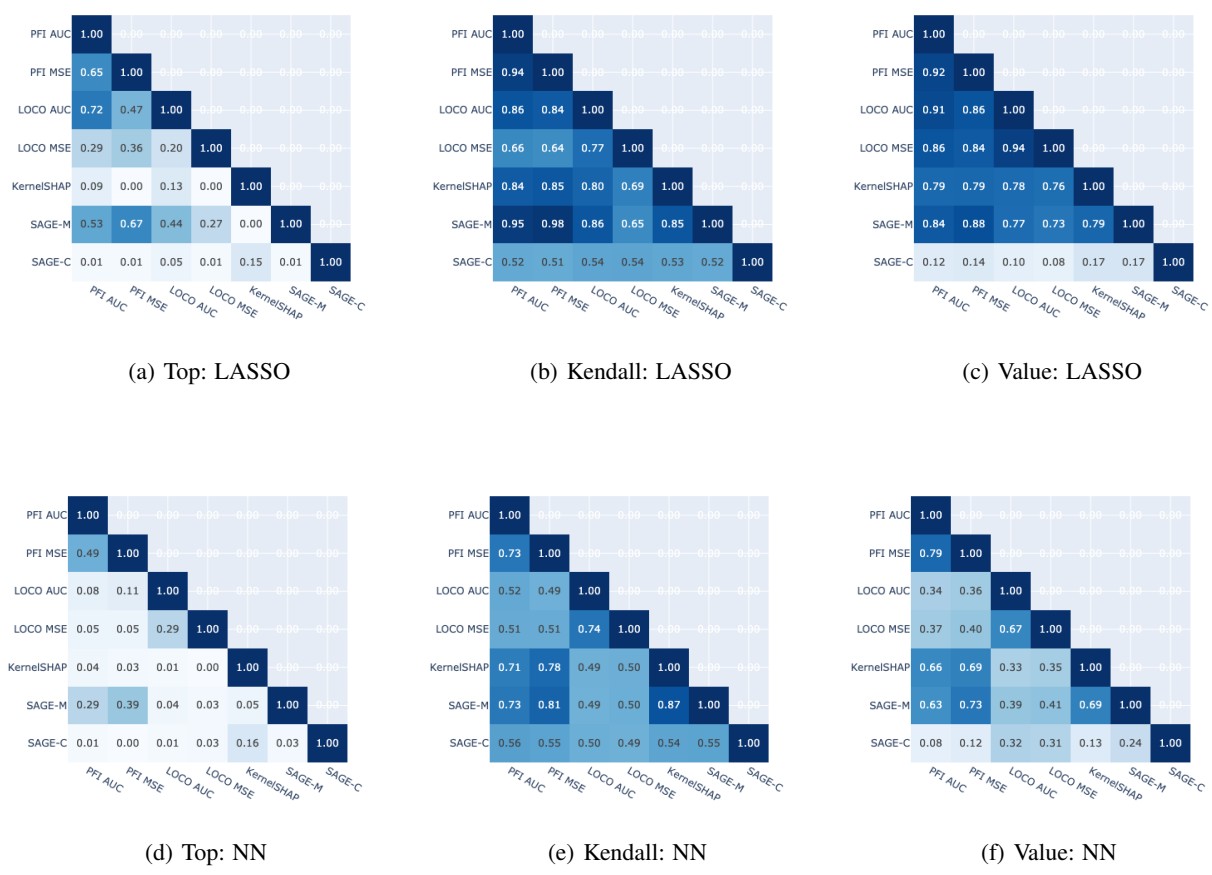

*Figure B.3.* Feature importance disagreement for LASSO and NN predicting Mortality in COPD (IPCI data). Disagreement is measured using the average over Top-5 feature, Feature ordering, and Feature value metrics over five repeats. Values closer to 1 indicate more agreement.

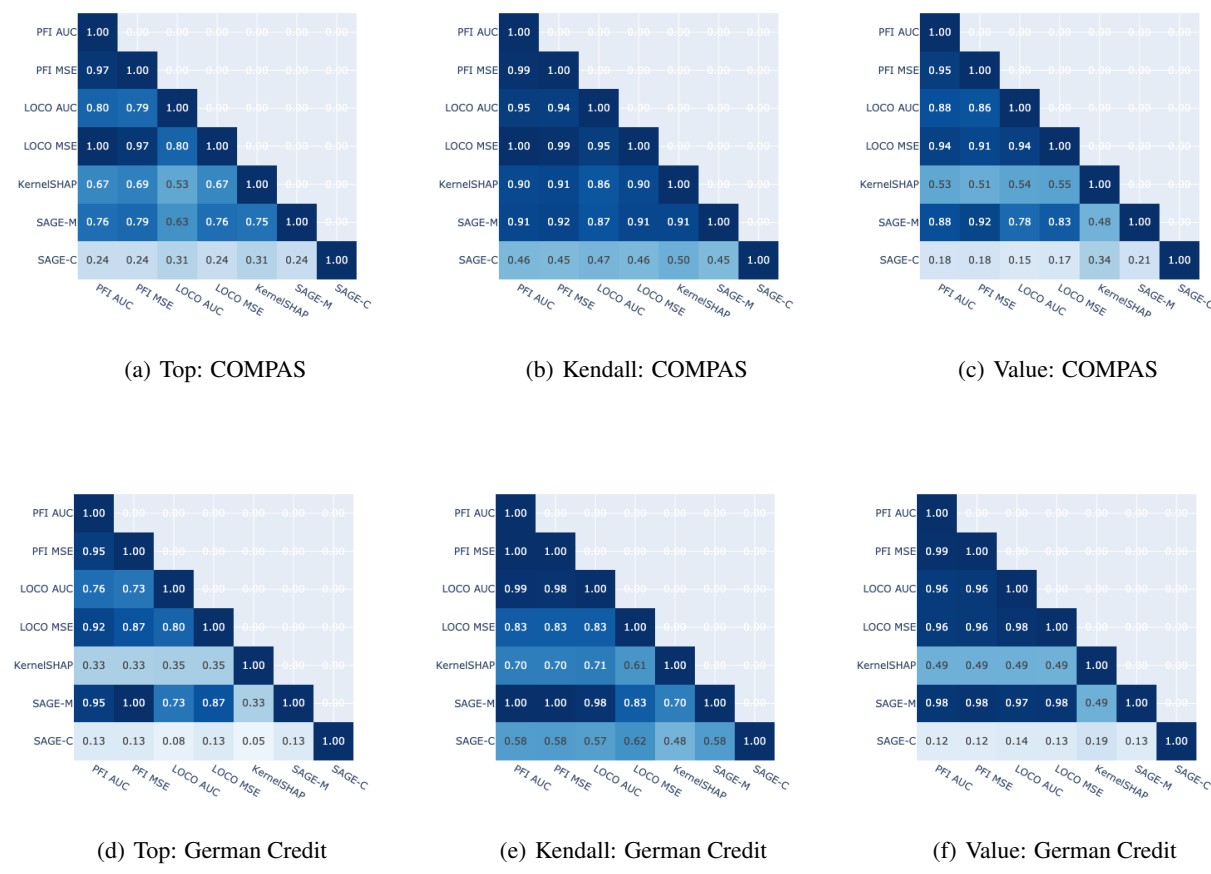

(a) Top: COMPAS     (b) Kendall: COMPAS     (c) Value: COMPAS

(d) Top: German Credit   (e) Kendall: German Credit   (f) Value: German Credit

*Figure B.4.* Feature importance disagreement for LASSO predicting COMPAS and German Credit (open-source data). Disagreement for all six metrics is averaged over five repeats. Values closer to 1 indicate more agreement.

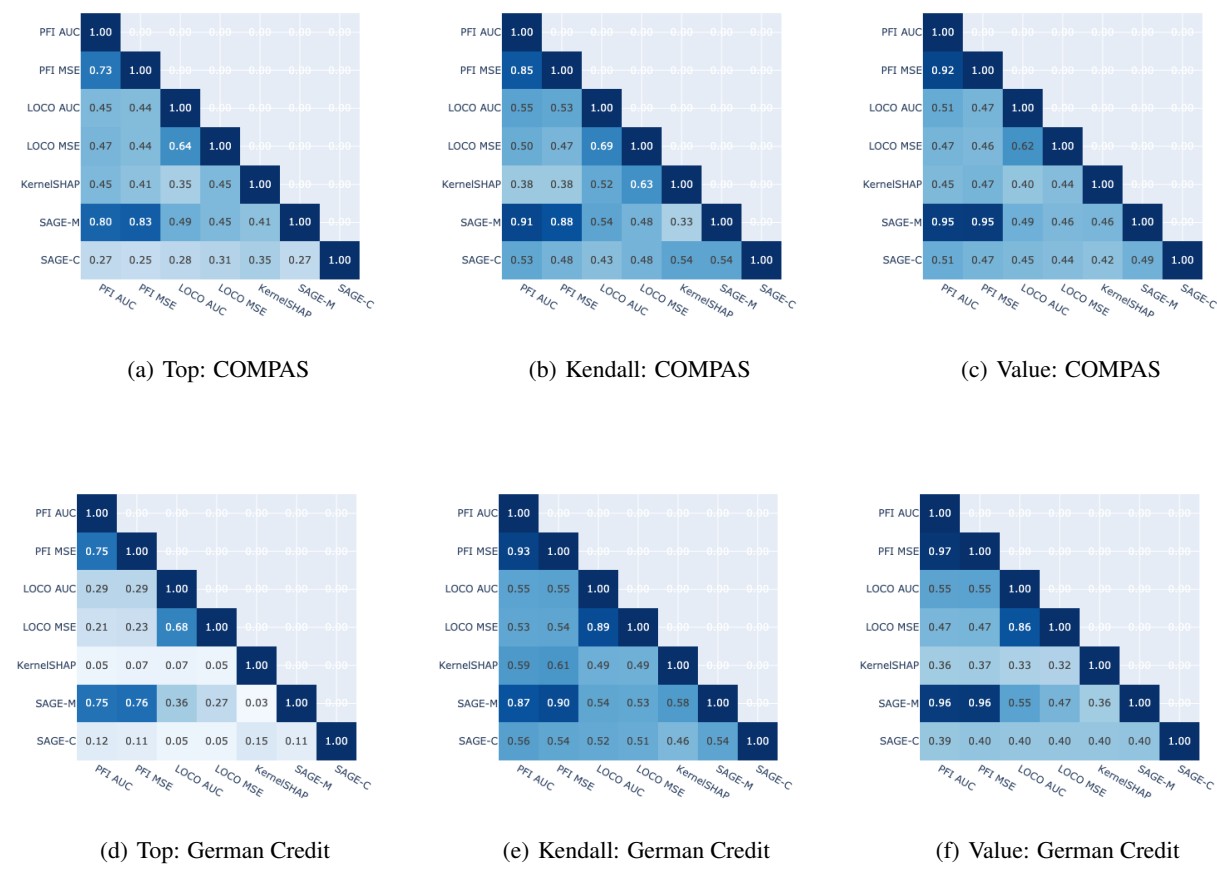

(a) Top: COMPAS      (b) Kendall: COMPAS      (c) Value: COMPAS

(d) Top: German Credit      (e) Kendall: German Credit      (f) Value: German Credit

*Figure B.5.* Feature importance disagreement for NN predicting COMPAS and German Credit (open-source data). Disagreement for all six metrics is averaged over five repeats. Values closer to 1 indicate more agreement.

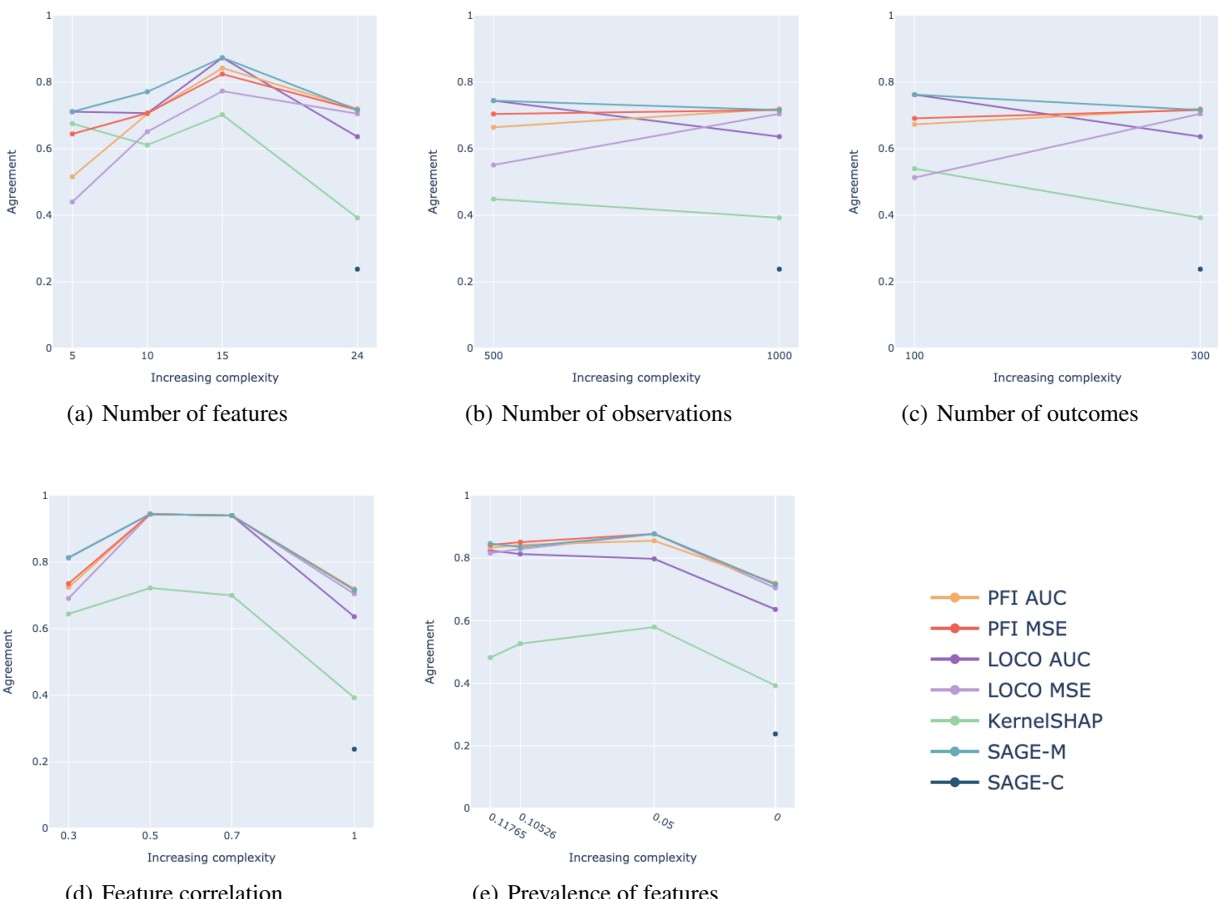

(a) Number of features

(b) Number of observations

(c) Number of outcomes

(d) Feature correlation

(e) Prevalence of features

*Figure B.6.* Understanding size of feature importance disagreement for varying elements of data complexity for the LASSO model for German Credit (open-source data). Disagreement is measured using the average over top-5 feature metrics (FA, SA, SRA) over five repeats. Values closer to 1 indicate more agreement.

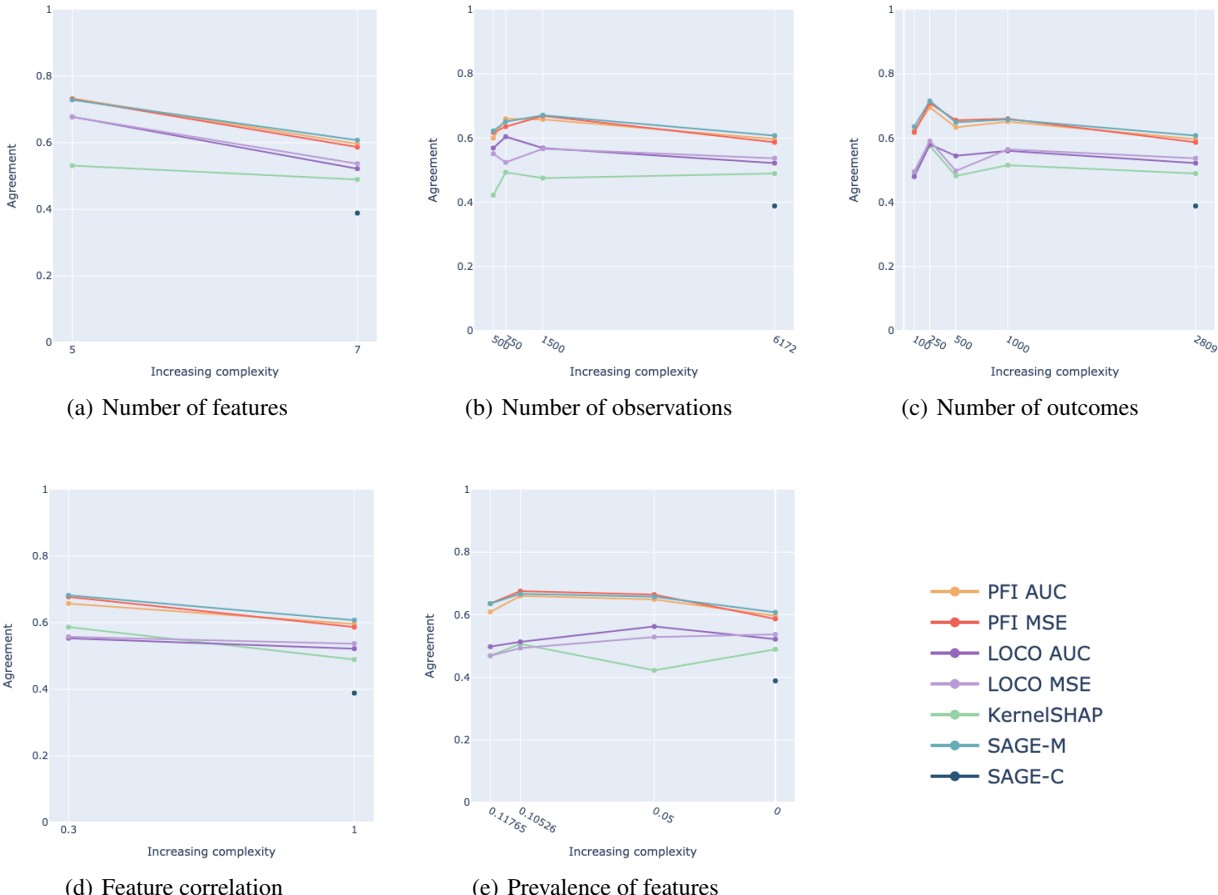

(a) Number of features

(b) Number of observations

(c) Number of outcomes

(d) Feature correlation

(e) Prevalence of features

*Figure B.7*. Understanding size of feature importance disagreement for varying elements of data complexity for the NN model for COMPAS (open-source data). Disagreement is measured using the average over top-5 feature metrics (FA, SA, SRA) over five repeats. Values closer to 1 indicate more agreement.

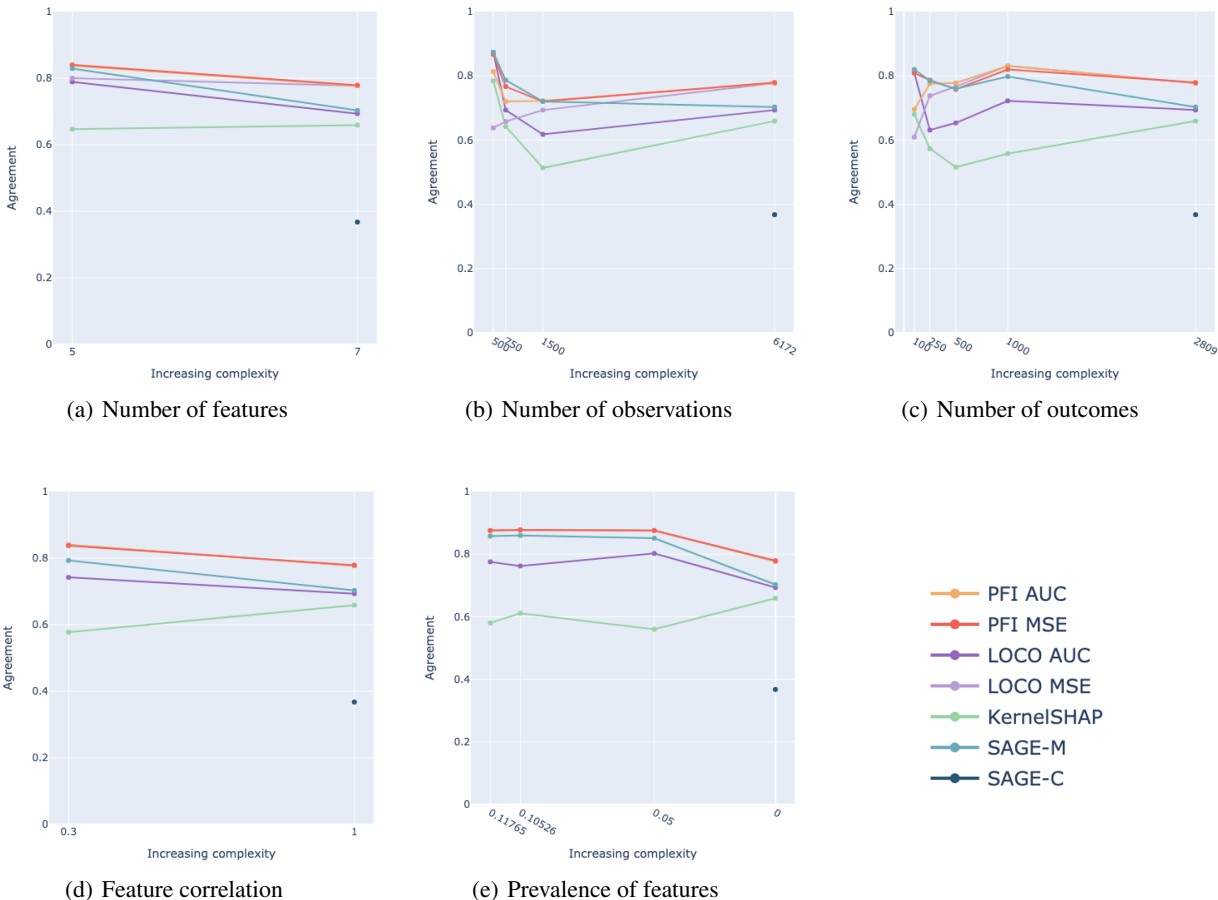

*Figure B.8.* Understanding size of feature importance disagreement for varying elements of data complexity for the LASSO model for COMPAS (open-source data). Disagreement is measured using the average over top-5 feature metrics (FA, SA, SRA) over five repeats. Values closer to 1 indicate more agreement.

