# OpenReview forum: "Understanding the Size of the Feature Importance Disagreement Problem in Real-World Data"
_ICML.cc/2023/Workshop/IMLH — IMLH 2023 Poster_

### Official Review · Reviewer_hcEx · 2023-06-11
**An important practical issue studied in this paper**

**Rating:** 7
**Confidence:** 3

**Review:**

In this paper, consistency of feature importance explanation is considered. This is an important aspect, if we want to truly trust a model because of interpretations provided, these interpretations should be consistent. Few metrics has been considered including both ranking and value based metrics on 4 datasets (2 publicly available). Results from 4 well-known methods are reported. I think these results could be insightful for community and hence, I suggest acceptance of the paper for the workshop.

---

### Official Review · Reviewer_jqDu · 2023-06-14
**The authors empirically analyze and understand the explanation disagreement problem and give reasonable explanations.**

**Rating:** 6
**Confidence:** 3

**Review:**

This paper is mostly well-written, though the notations could be improved (e.g., need to explain what $||$ means, and why it is $[]$ in $S_{d,m} \in [+, -]$ instead of {}). The authors propose a new framework to understand the explanation disagreement problem. In their experiment, two machine learning algorithms (simple linear L1 regularized model LASSO and more complex NN) and seven model-agnostic feature importance methods are considered on four datasets. The experiment conclusions are expected and the authors give reasonable explanations. It is an interesting paper but it is not original or significant in my perspective. Specifically,
1. Can the authors give more theoretical or intuitive explanations on the proposed evaluation framework (Figure 1)? This framework looks reasonable and understandable, and so lacks of originality. The authors need to demonstrate the significance of this newly proposed framework.
2. From line 134 to line 149 in the right column, the authors demonstrate the advantages of using real-world data. But what should we do if we do not know the truth? How do you know that the importance features that most of the methods agree on are true?
3. I am confused by the metrics to quantify the disagreement (line 184-205 in the right column). Should "$F_{K, u} \cup F_{K, v}$" be "$F_{K, u} \cap F_{K, v}$" in FA, SA and SRA, since the higher the values, the more agreement among the two methods. Moreover, why the larger the MAE/RMSE, the more agreement?
4. It would be more convincing to also try tree-based methods on tabular datasets other than LASSO and NN.

---

### Official Review · Reviewer_EAxS · 2023-06-19
**The paper deals with an interesting problem, but there are concerns about the proposed method and the experiment**

**Rating:** 5
**Confidence:** 4

**Review:**

The paper proposes an evaluation framework to measure the impact of data complexity on the size of the disagreement problem and investigates this problem in real-world datasets.

While the paper deals with an interesting problem, there are some concerns about the proposed evaluation method and the implementation:

1. There is concern about the key contribution - it may not be valid to directly compare the FI disagreement across different data complexity. For example, when the number of features is changed, the "baseline" disagreement across different methods do not remain the same (e.g., Top-5 feature agreement would be expected to be larger for a dataset composed of 10 features than 100 features, this is not due to any nature of FI methods, but only due to the definition of disagreement). It may need a careful analysis to decide what is a fair comparison across different data complexity (e.g., what would be an unbiased baseline for different complexity).

2. There are only two to four different complexities being tested in the experiment, and it may not be enough to summarize any trend.
In addition, a systematic evaluation on more datasets may be needed if any general conclusion is made.

---

### Meta-Review · Area_Chair_wCFV · 2023-06-19

**Recommendation:** Accept (Poster)
**Confidence:** 4

**Metareview:**

The work provide an empirical study on the disagreement of feature importance methods for given data and models. The overall experiment design and quality of the paper is good, although the study is less novel and original. The authors are encouraged to incorporate the reviewers' comments to improve the manuscript on the clarity of the notations.

---

### Decision · Program_Chairs · 2023-06-20

Accept (Poster)